# Impact of Sampling Frequency on Low-Cost PM Sensor Performance including Short-Term Temporal Events in High PM Environments

Gulshan Kumar [1,*], Prasannaa Kumar D. [2,*], Saran Raj [3], Jay Dhariwal [4], and Seshan Srirangarajan [5]

[1,2,3,4]Department of Design, Indian Institute of Technology Delhi, New Delhi 110016, India
[5]Department of Electrical Engineering, Indian Institute of Technology Delhi, New Delhi 110016, India

**Correspondence:** Jay Dhariwal (jay@design.iitd.ac.in)

**Abstract.** Low-cost sensors (LCS) for particulate matter (PM) monitoring have gained popularity due to their affordability, compact size, and low power requirements. These sensors typically offer the capability to collect data at sampling rates that can be adjusted according to the application. However, the effect of varying the sampling frequency on sensor performance has not been thoroughly examined. This study explores how variations in sampling frequency influence the performance of low-cost PM sensors and identifies some possible use cases for different sampling rates. During this study conducted over a one month period, data from five SPS30 sensors was collected at 15 second intervals and then aggregated into 5, 10, 15, 30, and 60 minutes intervals. During the study, hourly $PM_{2.5}$ levels ranged from 117 $\mu g/m^3$ to 303.3 $\mu g/m^3$, with significant diurnal variations, influenced by temperature and humidity. It was found that changes in sampling frequency had minimal impact on sensor performance, as evidenced by comparable linearity and error metrics across different sampling intervals. However, the study also revealed that short-lived plume events could be missed at lower sampling frequencies. This suggests that for monitoring gradual changes in $PM_{2.5}$ levels, higher sampling frequencies do not necessarily improve measurement accuracy, but are crucial for capturing transient events. This study underscores the importance of optimizing sampling frequency based on specific monitoring objectives and the need to balance power consumption with data resolution, particularly in remote or battery-based deployments.

## 1   Introduction

Low-cost sensors (LCS) for particulate matter (PM) monitoring have generated significant attention within the scientific community due to their affordability and performance (Liu et al., 2020; Giordano et al., 2021). Their compact design makes them well-suited for portable applications, while their affordability allows for widespread deployment, facilitating the establishment of spatio-temporal networks for high-resolution pollution data (Yi et al., 2015; Jiao et al., 2016; Zheng et al., 2019). The term "low-cost" can be subjective and requires clarification. In our study, we define affordability in the context of low-cost PM sensors based on their capital cost relative to regulatory-grade instruments. According to the United States Environmental Protection Agency (USEPA), low-cost PM sensors are formally characterized by a capital cost of less than USD 1000. Moreover, their low power consumption makes them suitable for standalone use, including scenarios where battery or solar power can be

utilized to power them (McKercher et al., 2017; Lung et al., 2020; Gulia et al., 2020; Das et al., 2022). These sensors serve diverse purposes, ranging from personal exposure monitoring (Helbig et al., 2021; Xi et al., 2022) to the creation of extensive data collection networks for ambient air pollution monitoring (Koehler and Peters, 2015; Jain et al., 2021). Additionally, the USEPA's acceptance of LCS for non-regulatory applications further underscores their utility (Duvall et al., 2021; Malings, 2024).

These LCS have the capability to record data at very high frequency, which typically translates to more number of samples per hour. For most LCS, the highest frequency may reach 1 sample per second (Sensirion, 2024; Plantower, 2024; Honeywell, 2024) and can be configured to lower frequencies. In the context of this study, high frequency corresponds to sampling intervals shorter than one hour, with the highest frequency corresponding to a sampling interval of 15 seconds, while low frequency corresponds to longer sub-hourly sampling intervals, up to one sample per hour. LCS are typically calibrated against optical particle counters (OPC) using known particles such as monodisperse polystyrene latex (PSL) beads for lab testing (Kaur and Kelly, 2023). The calibration involves averaging the high-frequency LCS data to match the sampling frequency of reference monitors, followed by assessment against the USEPA-recommended metrics (Duvall et al., 2021).

High-frequency data may capture short plume event and help in the identification of pollution hotspot and source identification, but configuring LCS to operate at high sampling rates leads to increased power consumption. This issue is particularly critical in remote deployments where a direct power supply may not be available, and it can rapidly drain battery resources. Conversely, lowering the sampling frequency may lead to the loss of critical information. Despite the widespread deployment of LCS, there remains a significant gap in research and guidelines regarding the optimal sampling frequency for balancing data resolution, power consumption, and performance alignment with the reference monitor.

In the context of short-term exposure measurement, accurately capturing brief, high-intensity plume events such as diesel generator emissions (Gilmore et al., 2006; Greim, 2019; Fadel et al., 2022), vehicular emissions (Lipfert and Wyzga, 2008; Zhang and Batterman, 2013), cooking (Balakrishnan et al., 2004; Sharma and Jain, 2019; Chu et al., 2021; Xiang et al., 2021), or waste burning (Wiedinmyer et al., 2014; Kumari et al., 2019) is very important. These transient spikes in $PM_{2.5}$ levels can have a significant impact on daily air quality and are linked to adverse health effects, including airway inflammation and other complications (Salvi et al., 1999; Gong et al., 2003; Pope and Dockery, 2006). Although both short- and long-term exposures are associated with health risks, the methods to monitor them are different. Long-term exposure can often be monitored with lower frequency sampling, which is adequate to detect general pollution trends. However, accurately assessing short-term exposure requires high-frequency data to capture the plume events. In personal exposure monitoring, it is important to evaluate both the exposure level and the corresponding dose (Borghi et al., 2020, 2021). Low sampling frequencies may suffice for broad exposure assessments, but high-frequency data is necessary to accurately measure the dose. This highlights the need to optimize sampling strategies based on specific monitoring objectives, such as assessing overall exposure or capturing the dose during short-lived events.

This study was conducted in New Delhi, India's capital, due to its severe air quality challenges. The annual average $PM_{2.5}$ concentration in New Delhi reached 100 $\mu$g/m$^3$ in 2021-2022, far exceeding the World Health Organization (WHO) guideline of 5 $\mu$g/m$^3$. The city's pollution issues are compounded by unfavorable meteorology, geography, and inward pollutant transport,

in addition to local emissions. The study was carried out during the month of October, a critical transition month when air quality begins to deteriorate, offering a diverse range of pollution levels. The Sensirion SPS30 sensor was selected based on its superior performance in our earlier collocated study in New Delhi (Kumar et al., 2025), aligning with recent research on sensor selection in heavily polluted environments. This setup allows us to evaluate the sensor's capabilities under challenging and dynamic air quality conditions. This study utilizes a Sensirion SPS30 unit collocated with a reference-grade beta attenuation monitor (BAM) over a one-month period. The SPS30 collected data at 15 seconds interval, which were then used to extract samples at the midpoint of each 5, 10, 15, 30, and 60-minute intervals, rather than averaging over these intervals. This approach was chosen to simulate the scenario where sensors are configured to sample data only once per interval, allowing us to assess the impact of different sampling frequencies. This setup enables us to evaluate how varying sampling intervals influence the performance of LCS compared to reference instruments and their ability to detect short-lived, local pollution events. Our analysis focuses on two key questions: (1) How do different sampling intervals affect LCS performance compared to the BAM reference instrument? and (2) What is the impact of sampling frequency on the detection of short-lived, local pollution events? By addressing these questions, we aim to optimize the use of LCS for both long-term air quality monitoring and for capturing transient pollution events in challenging urban environments such as New Delhi.

## 2 Methods

### 2.1 Sensirion SPS30 PM$_{2.5}$ Sensor

The SPS30 is a PM sensor from Sensirion and has been used in multiple low-cost monitors and well-evaluated for outdoor applications (Tryner et al., 2020; Roberts et al., 2022). Low-cost PM sensors estimate the PM mass concentration based on light scattering. SPS30 uses a 660 nm laser diode for light scattering (Sensirion, 2024). The sensor uses an auto-cleaning feature to maintain the airflow rate by cleaning the airflow path. The sensor also supports sleep mode for reducing power consumption when it is not actively sampling. The current drawn is as low as 50 $\mu$A during sleep mode (SPS30, 2020). Based on the coefficient of determination ($R^2$), for outdoor PM$_{2.5}$ measurements, SPS30 is among the best-performing LCS (AQSPEC, 2024). In this study, five Sensirion SPS30 sensors were deployed, and throughout this article, we refer to them as Sensirion units, each identified by their respective serial number (e.g., Sensirion-1, Sensirion-2, etc.)

### 2.2 Sensor Cluster Design

A custom monitoring device was designed and developed for this study using five SPS30 LCS (refer Supplementary Figure S1). In this device, SPS30 sensors communicate with the microcontroller (MCU) using the I$^2$C protocol. Each SPS30 sensor has a predetermined I$^2$C address, 0x69 (Sensirion, 2024). Since I$^2$C uses common lines and all five sensors have identical addresses, an I$^2$C multiplexer was used (Texas Instruments, 2024). A custom-designed printed circuit board (PCB) with inbuilt real-time clock (RTC), SD card connector, OLED display, and power management IC was used. ESP32 MCU was used for data acquisition from the sensors. The data was stored along with the timestamp on an SD card. A 12 second interval between sensor

data acquisition was established to synchronize with the I$^2$C multiplexer. The data stored on the SD card can be accessed using the file transfer protocol (FTP) over a Wi-Fi client, with a magnetic reed switch facilitating the activation of the FTP protocol during data transfer. 3D-printed parts were used as sensor holders and to guide the airflow along the inlet and outlet paths. Each sensor's sampling inlet and outlet were isolated from the others using a 3D-printed part. We sample the SPS30 sensors at a fixed sampling interval of 15 seconds, and data for other sampling intervals (5, 10, 15, 30, and 60 minutes) are derived by selecting the sample at the midpoint of each interval. This approach was implemented to maintain consistency in data analysis across different sampling intervals. The sensors sample throughout the experiment without any sleep mode, ensuring uninterrupted data collection over the measurement period.

## 2.3 Reference Monitor BAM

The BAM-1020 from Met One Instruments served as the reference monitor. Operating on the beta-ray attenuation technique, this monitor is approved by the USEPA for regulatory purposes. The BAM-1020 was equipped with a PM$_{2.5}$ very sharp cut cyclone (VSCC) to ensure precise size selection of fine particulate matter. The PM$_{2.5}$ data collected from the BAM is available at an hourly resolution. Additionally, temperature and humidity data from the BAM's ambient sensor were also recorded. Throughout the study, the BAM's error logs were regularly checked. BAM directly measures particulate mass by quantifying the attenuation of beta radiation through collected particles, with attenuation dependent solely on particle mass regardless of physical or chemical properties. This differs fundamentally from optical measurement techniques, which rely on light scattering or absorption properties that vary significantly with particle composition, size, and shape. This distinction is crucial for source identification studies, as BAM provides consistent mass measurements across diverse particle types (from combustion products to sea salt), while optical techniques may yield variable responses to different emission sources despite similar mass concentrations, potentially confounding source attribution efforts.

## 2.4 Site Description and Meterology

The experiment site was located inside the Indian Institute of Technology Delhi, New Delhi campus (28°32'37.5"N 77°11'31.1"E). The LCS cluster and BAM were kept within 0.5 m of each other on the roof of a single-storey building. Their inlet heights were set at 1 m from the roof surface, facilitating unobstructed airflow throughout the experiment. The experiment was conducted during the month of October 2023. During the experiment, ambient PM$_{2.5}$ levels ranged between 11.7 $\mu$g/m$^3$ to 303.3 $\mu$g/m$^3$ with an average of 84.4 $\mu$g/m$^3$. In addition, the temperature ranged from 16.6 °C to 36.6 °C, with an average of 25.49 °C, and the relative humidity (RH) levels were between 27% and 93% with an average value of 65%. The recorded time series data is shown in Figure 1, highlighting PM$_{2.5}$ levels, temperature, and relative humidity trends during the study period. Please refer to Supplementary Table S1 for a summary of descriptive statistics, including mean, minimum, maximum, and percentile for these parameters.

The experimental setup was strategically placed on the roof of a single-storey building (refer Figure 2) to ensure unrestricted airflow, with no nearby obstructions. Around the study site, a campus road is about 40 m away, while taller buildings which

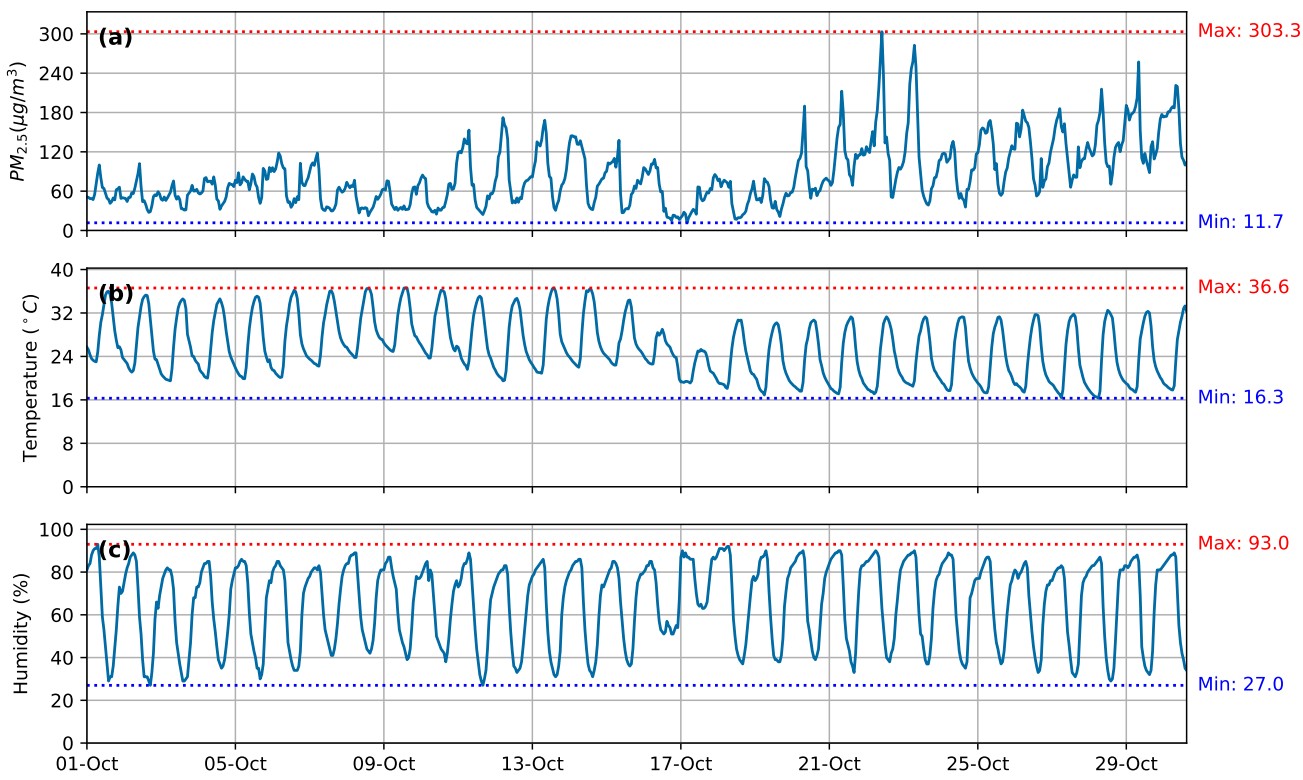

**Figure 1.** Time-series data of (a) PM$_{2.5}$ (BAM), (b) temperature, and (c) humidity. The minimum and maximum levels of each parameter are indicated for each subplot with blue and red dotted lines, respectively. Lower PM$_{2.5}$ levels are evident during the initial 16 days which can be attributed to higher temperatures, followed by higher PM$_{2.5}$ levels coinciding with lower temperatures, likely due to rain. The diurnal variation in humidity remained consistent, except on the 16th, 17th, and 18th of October, which saw rainfall.

are 4-5 storeys high, are located about 60 m away. In addition, a backup diesel generator (DG) is located about 30 m away at the ground level.

Since October falls in the post-monsoon season in India, occasional rainy days were observed during the experiment, notably impacting temperature trends following the rainfall on 16th and 17th October, 2023. Figure 1 shows the variations in meteorological conditions and PM$_{2.5}$ levels during the study period, highlighting the dependence between the environmental factors and air quality. It is observed that PM$_{2.5}$ levels increase with rising humidity levels in the morning. As temperatures rise in the afternoon, the humidity levels decrease, leading to a drop in PM$_{2.5}$ concentrations (refer Figure 3). In the evening hours, as humidity levels rise again, PM$_{2.5}$ levels gradually increase. The peak PM$_{2.5}$ levels are observed around 8:00 am and can be attributed to increased vehicular emissions, while lowest PM$_{2.5}$ levels are observed in the afternoon possibly due to higher temperature which may result in increased planetary boundary layer height and enhance ventilation. The diurnal variation in

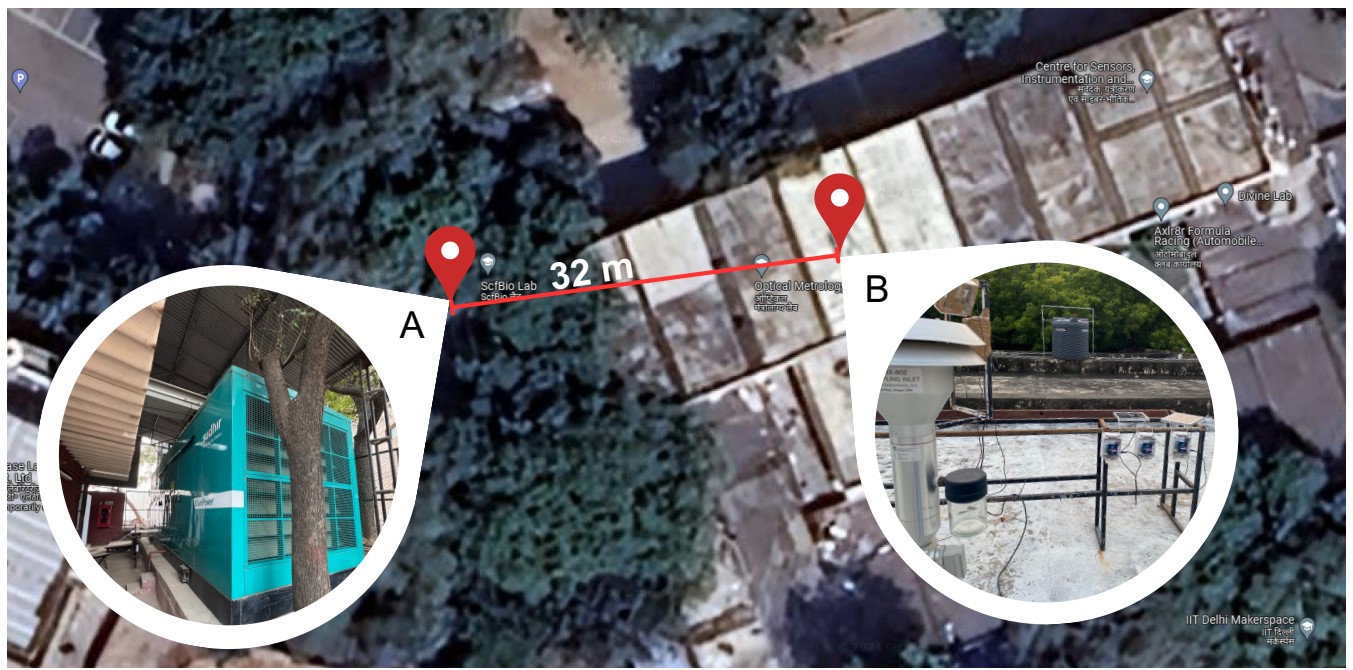

**Figure 2.** Location of study. The monitoring device was installed near a BAM on the rooftop of a building (B). Near the site there is a power backup building (A) located 32 m from B. Site A has three diesel generators (from Google Maps).

PM$_{2.5}$ levels ranged from $44.5$ $\mu$g/m$^3$ to $253$ $\mu$g/m$^3$, with an average of $107.35$ $\mu$g/m$^3$. On a large majority of the days $(83\%)$, diurnal variation was found to be between $50$ $\mu$g/m$^3$ and $150$ $\mu$g/m$^3$.

## 3 Results and Discussion

During the experiment, each LCS recorded data at 15 second resolution. For 5, 10, 15, 30, and 60-minute intervals, samples were extracted from the midpoint of each interval. Additionally, the 15 second resolution data are averaged to calculate hourly average. This distinction allows for a comparison between hourly averages derived from continuous high-resolution data and single-point samples taken at longer intervals. This results in six hourly average datasets for each LCS unit based on different sampling intervals. These hourly averaged LCS datasets were grouped by sampling interval and compared to BAM measurement. The time-series plot of this data reveals significant overlap among the LCS datasets indicating high precision. To quantify this precision, standard deviation (SD) and coefficient of variation (CV) are calculated in accordance with the USEPA guidelines (Duvall et al., 2021). Daily averages for all the LCS units were computed using 15 second resolution data over the 30 day study period. This month-long dataset allows for diverse ambient exposure conditions and diurnal variations. During

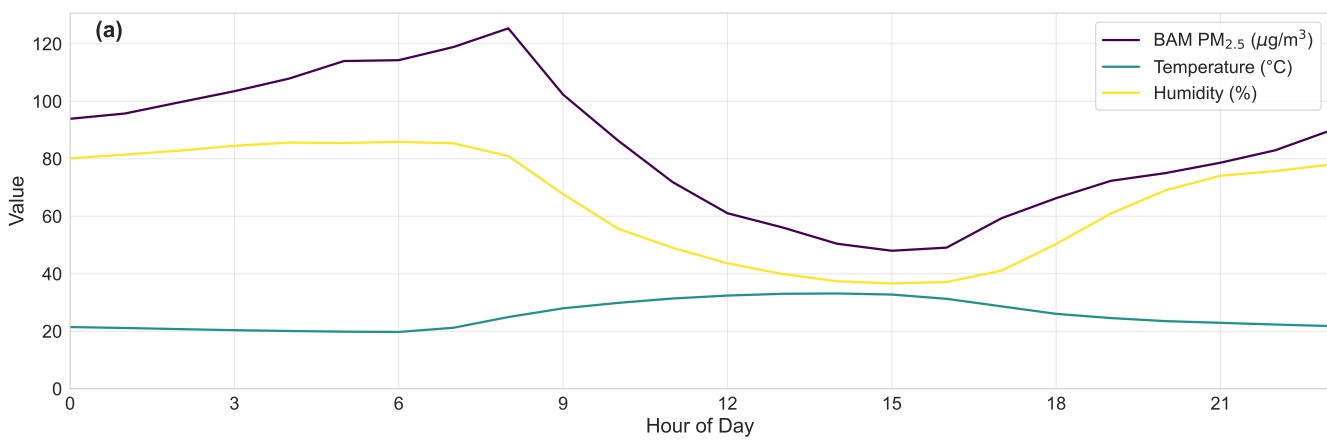

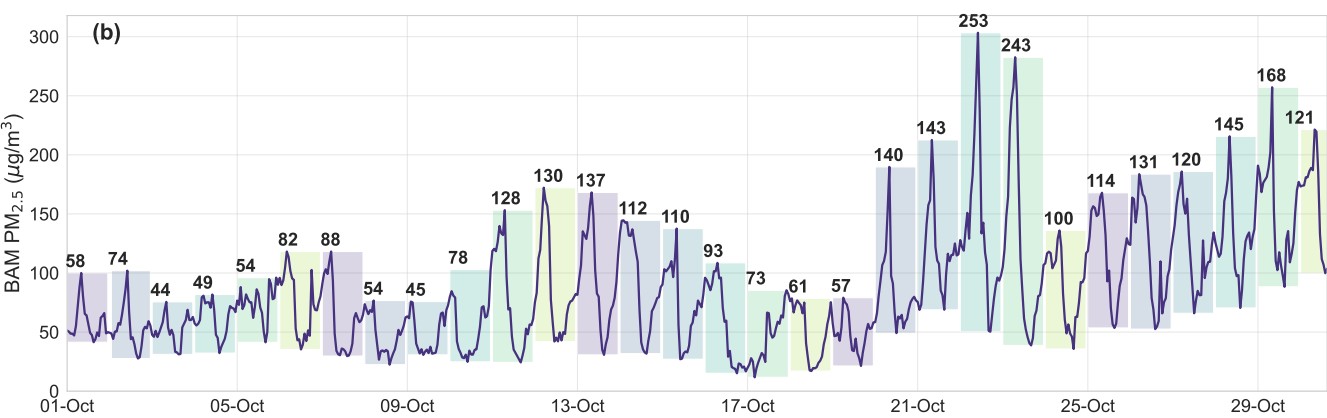

**Figure 3.** (a) Hourly averaged diurnal variations of PM$_{2.5}$ (BAM), temperature, and humidity levels. The x-axis denotes the hour of the day. PM$_{2.5}$ levels exhibit a direct correlation with humidity and an inverse relationship with temperature. Peak PM$_{2.5}$ levels are observed during morning hours with lower levels in the afternoon. (b) Time-series variation of PM$_{2.5}$ (BAM). Each day's diurnal variation is shown as a shaded region, with number at the top of each shaded region indicates the range of diurnal variation for that day.

|  | S1 (15 second) | S1 (5 min) | S1 (10 min) | S1 (15 min) | S1 (30 min) | S1 (60 min) |
|---|---|---|---|---|---|---|
| $R^2$ | 0.93 | 0.93 | 0.93 | 0.93 | 0.93 | 0.92 |
| Slope ($m$) | 1.38 | 1.38 | 1.38 | 1.38 | 1.38 | 1.39 |
| Intercept ($c$) | -31.63 | -31.48 | -31.81 | -31.50 | -32.04 | -32.42 |
| MAE ($\mu$g/m$^3$) | 18.10 | 18.10 | 18.16 | 18.17 | 18.40 | 18.70 |
| RMSE ($\mu$g/m$^3$) | 25.68 | 25.68 | 25.77 | 25.69 | 26.03 | 26.34 |
| NRMSE (%) | 8.81 | 8.81 | 8.84 | 8.81 | 8.93 | 9.03 |

**Table 1.** Performance metrics for different LCS sampling frequencies against hourly average BAM data.

the study, hourly averaged reference BAM measurements ranged from 11 to 303 $\mu$g/m$^3$, with a mean value of 84.41 $\mu$g/m$^3$.
Despite this wide PM$_{2.5}$ concentration range, the calculated standard deviation (SD) of 3.92 $\mu$g/m$^3$ (below the USEPA limit of $< 5$ $\mu$g/m$^3$) and coefficient of variation (CV) of 4.38% (significantly lower than the USEPA limit of $< 30\%$) indicates high precision among the five LCS units (Duvall et al., 2021). Given this high level of precision, a single LCS unit (Sensirion-1) was used for the subsequent analysis, as repeating the analysis with the other LCS units would yield similar results.

### 3.1 Effect of LCS Sampling Interval on Linearity and Error

To assess the impact of LCS sampling interval on the correlation of the LCS data with BAM, we compute the coefficient of determination (R$^2$), slope (m), intercept (b), mean absolute error (MAE), root mean square error (RMSE), and normalized root mean square error (NRMSE) (Duvall et al., 2021; Zimmerman, 2022). Scatter plots between different hourly averaged samples of Sensirion-1 and reference BAM are shown in Figure 4, with error and linearity parameters indicated within each subplot. Each subplot in Figure 4 corresponds to a different sampling interval (e.g., 15 minutes, 30 minutes, etc.), where hourly LCS averages are computed from the raw measurements within that hour. Despite the different number of data points per hour (e.g., four data points for 15 minutes interval, two data points for 30 minutes interval), each subplot contains 720 hourly points (30 days × 24 hours), resulting in the same temporal resolution. While it may be commonly assumed that higher sampling rate leads to a better correlation with the reference BAM, it is seen from Figure 4 and Table 1 that different sampling intervals result in similar errors and other parameters. The reason behind similar results for different sampling intervals is averaging of the data and gradual variation in PM$_{2.5}$ levels.

### 3.2 Effect of LCS Sampling Interval on Hourly Average

An investigation into extreme cases is conducted using LCS data sampled at 15 second intervals. Hourly initial-final change in PM$_{2.5}$ concentrations was calculated, along with the hourly extremum difference within each hour. Hourly initial-final change represents the difference between the first and the last samples taken within an hour and hourly extremum is the difference between the highest and lowest values recorded within the hour. Hourly extremum is especially helpful in detecting short lived plume events. During our study, minimum hourly initial-final change was observed at 2023-10-08 14:00:00 (0.03 $\mu$g/m$^3$),

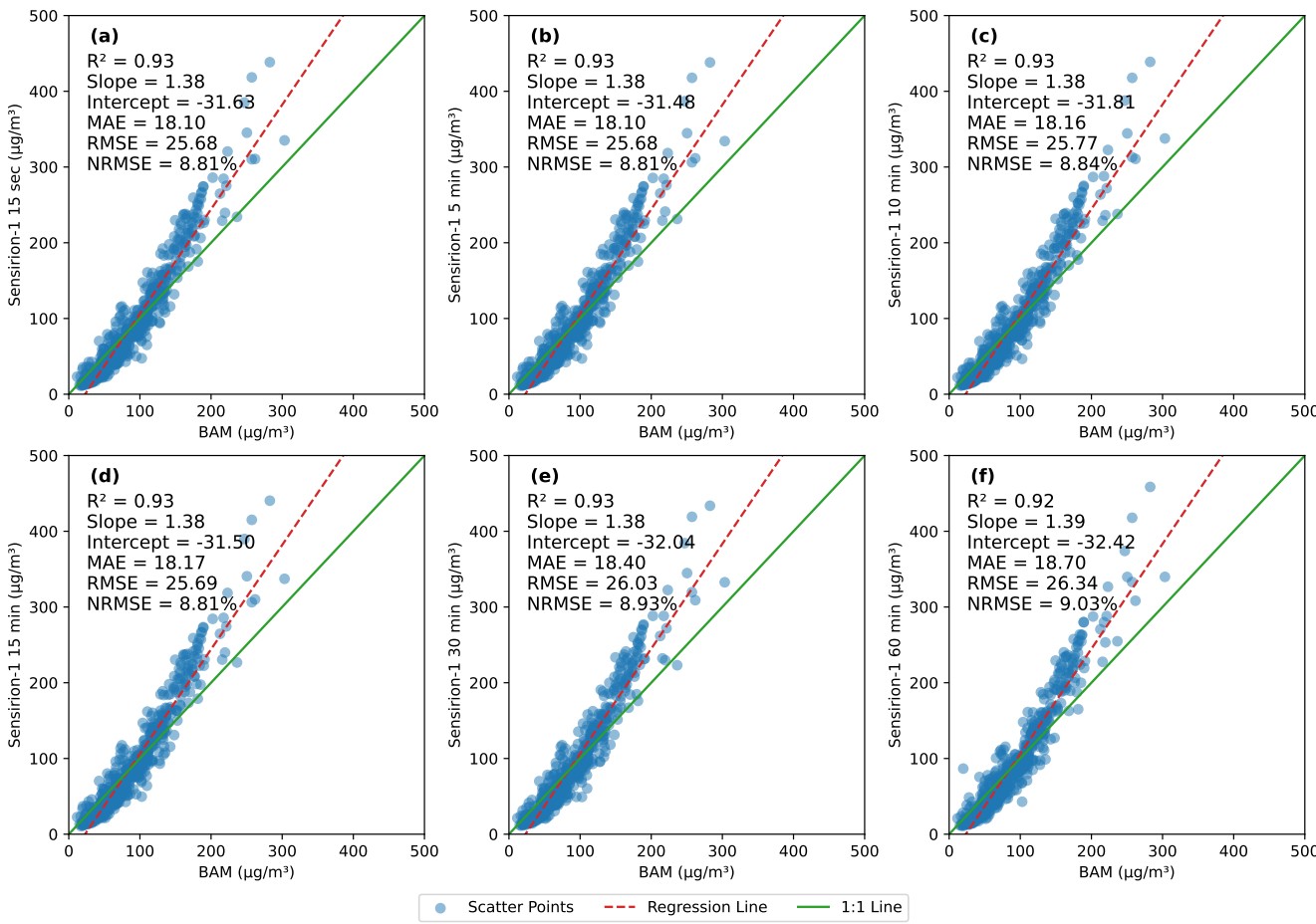

**Figure 4.** Scatter plots comparing LCS Sensirion-1 and BAM hourly averaged data for different LCS sampling frequencies. Each subplot includes a regression line in red and a 1:1 ratio line in green. Additionally, linearity and error parameters are indicated within each subplot.

while the maximum hourly initial-final change was observed at 2023-10-22 11:00:00 (179.035 $\mu$g/m$^3$) (refer Figure 5(a),(b)). Additionally, the minimum hourly extremum was recorded at 2023-10-18 12:00:00 (3.49 $\mu$g/m$^3$), while the maximum hourly extremum was observed at 2023-10-06 16:00:00 (351.69 $\mu$g/m$^3$) (refer Figure 5(c),(d)). Considering the hourly average of these extreme events shown in Figure 5 indicates that despite these being extreme events, the hourly averages for different sampling intervals remain consistent. This analysis shows that varying the sampling frequency does not significantly impact the hourly average.

### 3.3 Energy Consumption Across Sampling Frequencies

To evaluate the relationship between sampling frequency and energy consumption, we carry out an experiment using an Sensirion SPS30 sensor and an ESP32 microcontroller (MCU). The sensor-MCU system was configured to operate at six sampling

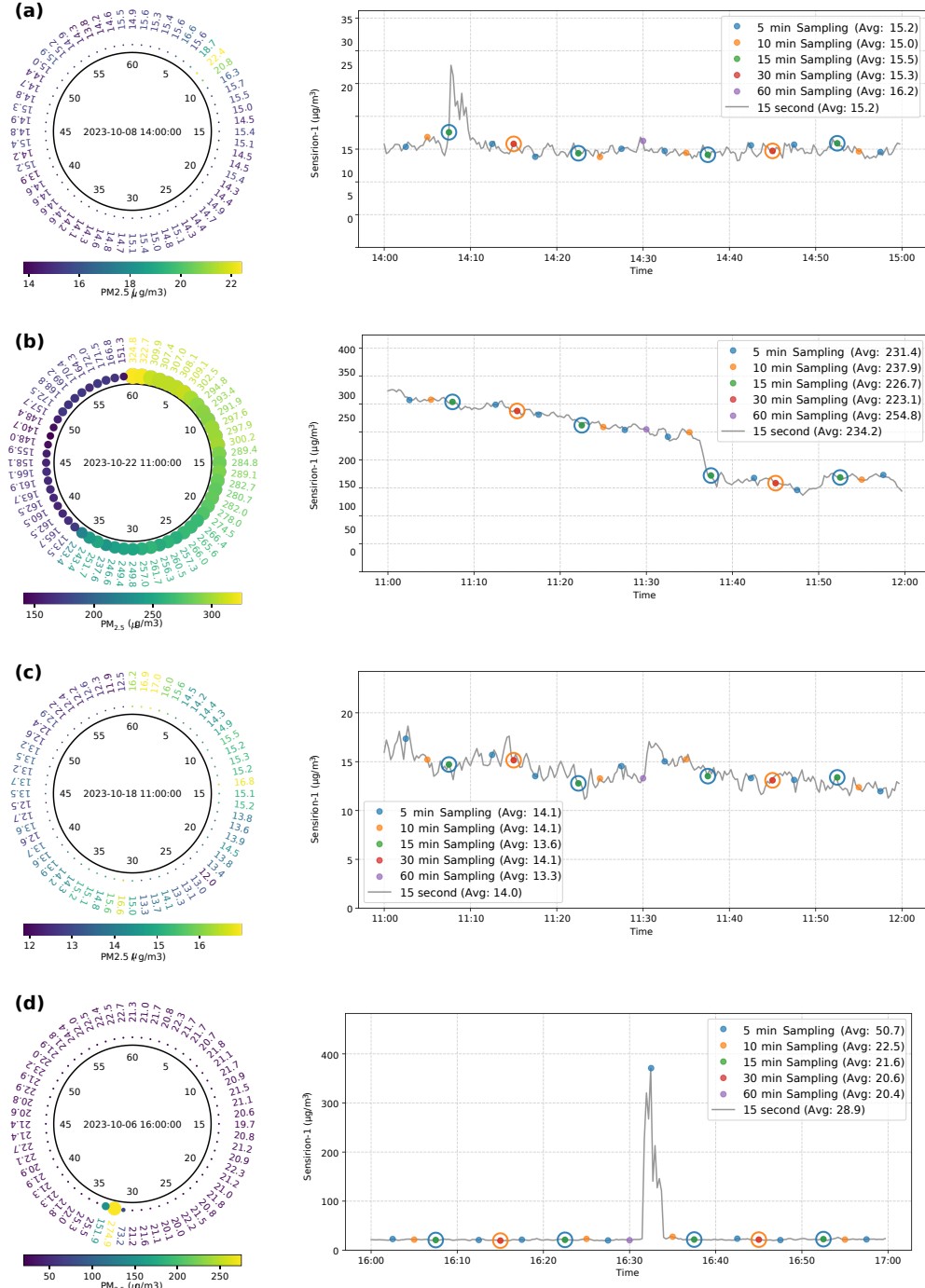

**Figure 5.** The clock to the left of each subplot displays PM$_{2.5}$ variation for a selected hour, with minute-averaged data around the circumference. The plot on the right within each subplot represents a time series plot showing 15-second data for the same hour, with colored dots representing different sampling intervals. The concentric circle represents that the two sampling intervals have the same sample. The four subplots highlight extreme cases: (a) minimum hourly change, (b) maximum hourly change, (c) minimum intra-hour variation, and (d) maximum intra-hour variation.

| S. No. | Sampling interval | Average current consumption (mA) |
|---|---|---|
| 1 | 15 seconds | 85.17 mA |
| 2 | 5 min | 3.97 mA |
| 3 | 10 min | 2.28 mA |
| 4 | 15 min | 1.72 mA |
| 5 | 30 min | 1.15 mA |
| 6 | 60 min | 1.03 mA |

**Table 2.** Average current consumption of the sensor and the MCU at different sampling frequencies.

intervals: 15 seconds, 5 minutes, 10 minutes, 15 minutes, 30 minutes, and 60 minutes. A power profiler kit 2 (PPK2) was configured in source mode to study the current consumption of the setup (Nordic Semiconductor, 2024). The SPS30 operates at 5 volts while the MCU functions at 3.3 volts. During non-sampling times, ESP32 enters deep sleep mode, drawing approximately 50 $\mu$A, while during active sampling current consumption is 80–100 mA. Table 2 shows that an inverse relationship exists between sampling frequency and current consumption. Reducing the sampling frequency, by increasing the sampling interval from 15 seconds to 60 minutes, decreased the current consumption by $98.6\%$. These results confirm that lowering the sampling frequency can significantly extend operational life in battery- or solar-powered deployments. While transient plume events require high-frequency sampling (discussed in Section 3.4), applications prioritizing trend analysis over event detection can adopt reduced sampling rates to minimize energy consumption. This trade-off is critical for remote deployments where energy efficiency can determine deployment feasibility.

### 3.4 Plume Event Detection

Plume events, characterized by short duration spikes in particulate matter concentration, can arise from various sources such as vehicular traffic, cooking, waste burning, and backup power generators. These events are crucial to measure as they often go undetected, when data is recorded at lower sampling rates, but can have significant implications. In this study, the potential applications of high-frequency PM data are examined based on the observation that sampling frequency has minimal impact on hourly averages. Data from LCS Sensirion-1 collected at 15 seconds interval, along with the corresponding hourly averages and the reference BAM hourly average data are analyzed to understand their differences (refer Figure 6). In Figure 5, the time series plot on the right side of each subplot shows 15 seconds data, and different colored dots within the time series plot indicate the samples taken for different sampling frequencies. Numerous spikes are observed in the 15 seconds sampled data that are not seen in the hourly averages. An example of such a spike is shown in Figure 5(d) and Figure 6 for the datetime 2023-10-6 16:00:00. Figure 6 shows the time series plot comparing the reference BAM (hourly data) and LCS S1 (15 seconds and hourly data) for $PM_{2.5}$. The diurnal variation in PM levels, driven by changes in RH, were presented and discussed in Figure 3. Figure 6 reveals plume events that were not discernible with hourly averages. These short-lived events, lasting less than 5 minutes, are highlighted with green arrows in Figure 6 but have minimal impact on hourly averaged data. Such events occurred on 13 out of

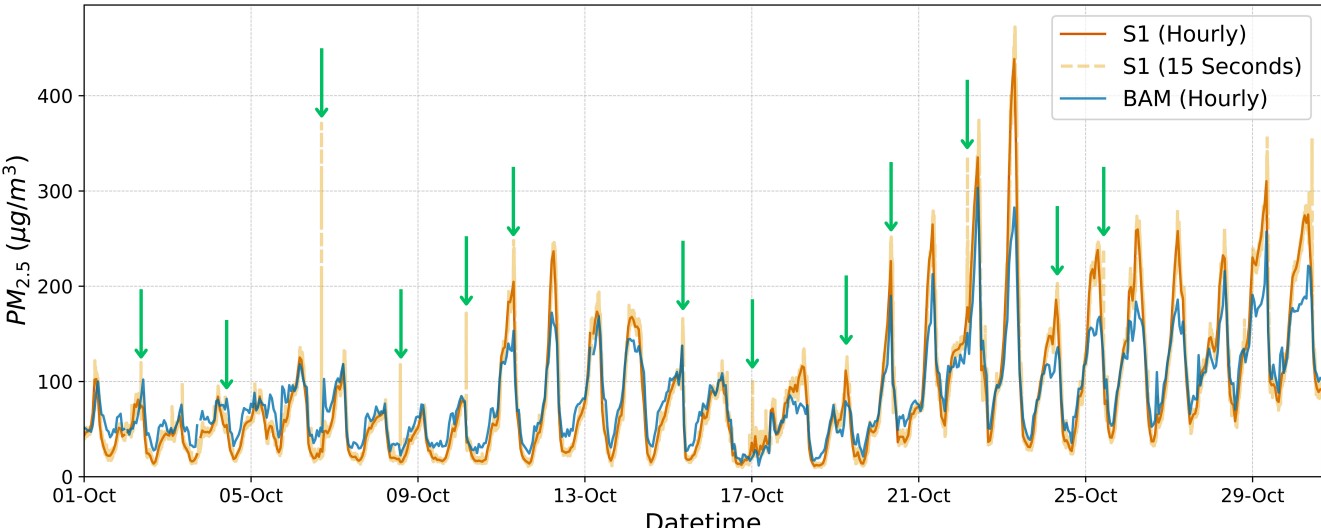

**Figure 6.** Time-series plot of 15 seconds and hourly averaged $PM_{2.5}$ data from the LCS Sensirion-1 and BAM. The hourly averages of BAM and S1 exhibit a consistent trend, similar to the 15 second data from Sensirion-1. However, the 15 second data of Sensirion-1 reveals additional spikes (indicated by green arrows), corresponding to plume events. These transient events are not captured by the hourly averaged data from S1 or the hourly data from BAM.

30 days during our experiment. In our investigation, we identified a backup power generator located approximately 32 m from the experiment site as a significant source for these plume events. Emissions from diesel generators, particularly during startup and operation, are well documented in the literature (Zhu et al., 2009). Analysis of the generator's operational log revealed that its use during power outages was responsible for 61% of the observed spike events. The generator was operated for very short periods, producing plumes upon startup. Additional plume events could potentially be attributed to cleaning activities and other short-lived emission sources in the vicinity.

## 4 Conclusion

This study, conducted over a period of one month, where LCS were collocated with a reference BAM has provided several important insights into LCS performance. The LCS and reference BAM exhibit consistent trends, along with a correlation between humidity and $PM_{2.5}$ levels. Higher $PM_{2.5}$ levels were observed in the morning hours due to higher humidity and vehicular activity, while lower $PM_{2.5}$ levels were recorded in the afternoon as the temperatures rise. The SPS30 LCS units demonstrated high precision, with a CV of 4.38% during the 30 days study period. While previous studies have established the health risks associated with short and long-term $PM_{2.5}$ exposure, in this work we investigated the impact of sampling frequency on the ability to detect short-lived plume events. We showed that while lower sampling frequencies are adequate for monitoring long-term trends, higher frequency data is necessary to capture short-lived plume events. In addition, our analysis

revealed that varying the sampling frequency had minimal impact on the hourly measurement accuracy, however only high frequency sampling (15 seconds sampling interval) was effective in capturing the transient plume events. Although short-term health assessment was not within the scope of this study, the findings offer valuable guidance for future research, particularly in deciding the sampling frequency for different monitoring objectives. Additionally, our study suggests that for deployments based on low-cost, standalone PM monitors which need to be solar-powered or battery-operated, if the goal is to capture overall

pollution trend rather than short-lived events, lower sampling frequencies provide similar long-term averages as high-frequency data. This enables the PM monitors to minimize their energy consumption and extend their operational life thus allowing for their long-term deployment in remote or resource-constrained areas. This work underscores the importance of customized sampling strategy while validating the use of LCS for capturing long-term trends as well as short-term exposure events in air quality monitoring.

*Code and data availability.* The raw data is available at https://zenodo.org/records/14230696 (Kumar et al.). The code can be provided upon request from the author.

*Author contributions.* G.K. carried out the data analysis and prepared first draft of the manuscript. P.K.D. developed the sensing monitor and carried out the experiments. J.D. and S.S. initiated this research, contributed to the design and planning of the experiments, helped with statistical analyses, provided feedback. All co-authors contributed to revising the manuscript.

*Competing interests.* The authors declare no competing interest.

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
