# Peer review of "Impact of Sampling Frequency on Low-Cost PM Sensor Performance including Short-Term Temporal Events in High PM Environments"

_Aerosol Research, 2024_

## Referee Comment (RC1)

**Review of "Impact of Sampling Frequency on Low-Cost PM Sensor Performance"**

**General Comments**

This manuscript investigates the impact of sampling frequency on lower-cost PM sensor precision, accuracy, and plume detection capability in a polluted megacity, Delhi. The dataset itself could be a useful training/exploration dataset with the right application in mind. It spans a dynamic polluted month with variable meteorological conditions while collocated with a reference instrument. However, the paper is largely unfocused on a specific application - reading as an unhappy hybrid of a measurement report (listing off communication protocols and daily readings), and letters paper (brief punchy analysis). As a result, the findings are muddled with distracting information more suitable to an Supplement. Furthermore, the analysis itself lacks impact. The findings on LCS sensor accuracy as a function of averaging time has been investigated previously in literature - in North India no less (doi.org/10.5194/amt-11-4823-2018) - however, here it lacks a nuanced discussion of aerosol optical characteristics and the influence of different sensor technology on the underlying quantification of plumes. It is unclear how these findings could be directly applied in a novel way for health studies, regulation, source apportionment, etc.

In addition to the underlying structural problems of the manuscript, the organization of the paper is weak. Nearly every figure requires regeneration or rethinking to meet the standards laid out by the publisher. Furthermore, literature review largely sticks to well cited more general work (e.g., Zimmerman et al) while not citing similar work from the region (e.g., doi.org/10.3390/s20051347). A more through analysis of work from Delhi and similar high background signal environments is sorely needed to better contextualize the results.

With these criticisms in mind, as well as my more specific comments below, I recommend the paper be **rejected** as unsuitable for publication in AR at this time. I empathize with the authors as I fully understand the difficult and often grueling nature of field work and scientific writing. I believe the underlying data is useful. If the authors develop a new hypothesis focusing on a specific application (perhaps in coordination with the other instruments/historical data at IIT), the work could eventually lead to a useful reference for the broader community. I wish all the authors good luck.

**Specific Comments**

- The title is too vague. I recommend a hypothesis centered title, something like "Lower Cost PM Sensor Accurately Quantify Short-term Plumes in High Background Polluted

Environments." This is more informative to readers and easier to find in search engines.

- Citation formatting is incorrect. In-text citations should be in chronological order, and the references section should be organized alphabetically by last name. This may seem overly picky, however cross checking citations is key to the review process, and missing this step unnecessarily added time to my review. See other recent preprints (e.g., ar.copernicus.org/preprints/ar-2024-35/) for examples.

- In the introduction, please define what affordability means in the context of "low-cost" sensors. Are you using a benchmark for "low-cost" (e.g., compared to other non-regulatory optical monitors)? Furthermore, it maybe worth noting these are low capital cost sensors - clearly the work in deployment, and calibrating amounts to a very high cost.

- In the second paragraph of the introduction, you write LCS are compared to OPC. However, recent work (doi.org/10.1080/02786826.2023.2285935) has conclusively demonstrated LCS such as the Plantower cited this work are OPCs, albeit low efficiency OPCs. Please briefly clarify the differences in operating methodology between a reference OPC and an LCS OPC. Additionally, highlight how this difference could lead to mischaracterization of short term peaks. Especially since OPCs infer mass concentration from particle counts rather than directly measuring mass.

- The third paragraph references opinions or beliefs from the scientific community using language like "widely assumed" or "lack of understanding" but offers no literature or evidence. Please either support these statements with some theoretical backing in supplement, and/or cite (preferably recent) review work on the current understanding of high resolution monitoring.

- Furthermore, in the third paragraph of the introduction it would help to be specific when using terms like "high frequency" and "low frequency." Techniques such as eddy covariance would consider 1 Hz or higher standard, while many epidemiological studies use annual or decadal data as standard. Only "high frequency" is defined earlier in the manuscript.

- Again, in paragraph 4 of the introduction it is worth discussing what the LCS actually measures. For example, most PM LCS don't accurately identify resuspended dust from vehicles which can make up a significant portion of supramicron (dp > 1 micron) PM2.5 nor fresh vehicle emissions (dp < 0.1 microns). While the effective size range of most PM LCS is still useful, it largely lies in the accumulation mode, which includes a big share of the regional aerosol burden rather than the hyperlocal burden. Please rework this paragraph to discuss which short-term PM signals LCS are useful to quantify in the light of previous epidemiological work and aerosol fundamentals.

- The last paragraph of the introduction needs to be more specific on why this month was useful, why the location is valuable, and why the Sensiron sensor rather than the Plantower, or Honeywell monitor. Citing other work contrasting the Sensirion with other sensor models, especially in Delhi or similar environments, is useful.

- There is a key strength of this manuscript which is excluded from the introduction and should be better described. This work focuses on characterizing plumes in highly polluted ambient environments. Much of the previous work in this area focus on comparing plumes in low to moderately polluted ambient environments in which the signal to background is high so analysis is straightforward. Despite the largely regional signal in Delhi, for which the design of many LCS is optimized, this work demonstrates its usefulness in source identification.

- Figure 1 should go to the Supplement, as should probably all of Section 2.2. These details are ubiquitous to most LCS packages, and are distracting in the main manuscript.

- I suggest adding 2 or at most 3 sentences in Section 2.3 contrasting how the BAM technique assess particle mass concentrations differently than an optical technique and why it matters for source identification.

- Lines 85-90 would be more efficiently communicated as a table with the 25th, 50th, and 75th percentiles for the relevant parameters.

- Figure 2 should be more clearly illustrated. For PM2.5 and RH, please ensure the y-axis minimum is **exactly** 0 bounded. For all plots the y-axis minimums and maximums should be clearly marked. The x-axis date labels should be more clearly marked, Oct 29 and Nov 1 overlap in a confusing to read way. Consider excluding the year after the first date marker. Adding grid lines would also improve interpretation.

- Figure 4 needs to be entirely regenerated. The top panel (a), should not contain both reasonably 0 bounded elements (i.e., PM2.5 and RH) on the same axis as temperature. Additionally, please label the beginning and end points on x-axis. Furthermore, when comparing different colored lines on the same plot, it is best practice to use color-blind friendly color schemes (see EGU publishing guidelines for more details) and/or different line styles. In this case, I'd recommend 3 different subplots for each variable. Panel (b), as in Figure 2 should have clear bounds, and be 0 bound on the y-axis minimum. Also, the red text in panel b should be better spaced to avoid overlap with the highlighted boxes or other text. Panel (c) looks to me like a screenshot from Plotly. I strongly recommend against Plotly and similar interactive tools for publication since they are difficult to reproduce. Also the y-axis label in Panel c is insufficient (no units!). Please remove Panel (c), or replace with a clearly labeled static boxplot, or use a table to describe the key percentiles of the monthly distribution.

- Figure 5 needs to be entirely regenerated. All scatterplots should be "square" - or equal ranges on both the x- and y-axes. The long lists of metrics would be easier to read on an accompanying table, with the panel labels in the upper left corner of the plot. Additionally, a legend denoting what each line and scatter point corresponds to is compulsory. To my eye, it appears as though there is some red shading around the regression line. Please explain how this is derived and how it is relevant. If not relevant, remove. I would guess its a confidence interval based on the sample size and therefore irrelevant and distracting.

- Figure 6 needs to be entirely rethought. As it stands the figure is too low resolution to read normally, I had to zoom in to read it and it rendered blurry. The "clock plots" don't follow best practices as highlighted in the previous figure comments, and don't add anything not clearly visible from the time series plots. Focusing on a few episodes can be helpful, especially as these demonstrate clear contrast, but the takeaway seems uninteresting.

- Figure 7 is also too low resolution (i.e., blurry) and should use colorblind friendly perceptually uniform colormaps in accordance with EGU publishing guidelines. Furthermore, there should be some discussion of aerosol properties which explain the "swings" in LCS hourly and sub-hourly data. Clearly the LCS is also systemically **underestimating** in addition to overestimating (probably due to RH?). Therefore, the magnitude of the plume spikes maybe incorrectly quantified relative to the BAM. Some baseline-spike decomposition algorithm could more holistically address these limitations. There are many in literature, especially in mobile monitoring.

- Overall the plume discussion doesn't focus on investigating aerosol characteristics. Simply finding a spike does not necessarily offer useful or actionable information. Therefore topics of interest to the community including background-plume decomposition analysis - which is key to both epidemiology and source characterization - are not clearly discussed or analyzed in the conclusion.

**Technical Comments**

- Line 65 - Needs a citation (maybe more than 1) when you assert its "among the best"
- Line 66 - It is customary to refer to Supplement items as Table S1, Figure S1, or Section S1. Refering to the sensors as S1, S2, etc. could be confusing for readers in publication.
- Lines 87-88 - This is not PM2.5 exposure, its simply the mean mass concentration.
- Line 101 - While higher temperatures *can* result in lower PM2.5 concentrations, I think this is merely an indicator of increased ventilation due to higher planetary boundary layer height during midday rather than a direct temperature effect - please clarify.
- Line 112 - This looks like the standard way to calculate SD, please either simply cite a statistics textbook/manual or move to supplement.
- Line 118 - Since you're using US-EPA standards, please state the US-EPA guideline for CV (about 0.1 or 10%, although 0.2 or 20% is a commonly used more lax requirement for LCS).
- Line 125 - Please explain or cite why this is the case

---

## Referee Comment (RC2)

**Title: Impact of Sampling Frequency on Low-Cost PM Sensor Performance**

The paper presents a field study in which a Low Cost Sensor measurement station for PM2.5 is designed and operated during one month on the roof of a building of Indian Institute of Technology (New Delhi campus). The data are analyzed and compared to reference measurement obtained by BAM Beta attenuation mass monitor thank to different sampling frequencies by the Low Cost Sensor Station. The general context of the study is interesting, it deals with configuration of sampling frequency of Low Cost Sensors regarding power consumption especially for remote deployments and what is it possible to characterize with in term of short pollution event. The precise objectives of the paper are clearly described.
The paper is well written, and results are clearly presented. It is in line with topics of *Aerosol Research*.
Nevertheless, some important points have to be accounted to improve the paper and avoid any misinterpretation.

**General comments:**

The main comment I have on the paper is to clarify the definition of the sampling frequency/sampling interval and related discussion on the effect of this parameter on the results. It is not clear to what correspond exactly LCS sampling frequencies named 5, 10, 15, 30, 60 min and how they are obtained.
As it is written it let thinking that data corresponding to such frequencies are obtained by doing periodic average on the raw measurements done by LCS working at an effective sampling frequency of 15 seconds. This means that sampling frequency of the LCS is not changed during experiments. This as to be clarified in the paper and the title of the paper should be adapted. In fact, if the frequency studied by the authors is a periodic average obtained by post-treatment it has no relationship with LCS intrinsic performance. The title should avoid such misunderstanding.
The authors should improve the paper by better describing how the LCS data are acquired: if it is always active sampling during one month of if there sleep mode periods between measurements periods?

**Specific comments**

**Page 3, line 80**
Precise/confirm that BAM unit is equipped with PM10 Inlet + PM2.5 Cyclone (which model VSCC or URG?)

**Page 5, lines 106-107**
Give additional information to explain the difference between data aggregated on 60 min interval and the hourly average.

**Page 11, fig. 7**
Improve readability of titles

**Page 11, lines 168-171**
The conclusion of the paper should be adapted to avoid misunderstanding about energy consumption minimization of LCS according to finding of this study. Energy consumption is

not directly studied here and no evidence are given that operation of LCS with lower energy consumption due to lower effective sampling frequency provide comparable measurements.

---

## Author Comment (AC1)

**Response to Reviewer 1 Comments on Manuscript ar-2024-39 "Impact of Sampling Frequency on Low-Cost PM Sensor Performance"**

The authors would like to thank the editor and reviewers for their valuable feedback on the manuscript. In this document, we present our responses to the reviewer comments and suitable changes will be made in the revised version of the manuscript addressing these comments. For the reviewers' convenience, the reviewer comments are shown in **black**, and our response to these comments are shown in **blue**
* * *
**Reviewer 1**

**General Comments**

**Reviewer Comment 1.1** — This manuscript investigates the impact of sampling frequency on lower-cost PM sensor precision, accuracy, and plume detection capability in a polluted megacity, Delhi. The dataset itself could be a useful training/exploration dataset with the right application in mind. It spans a dynamic polluted month with variable meteorological conditions while collocated with a reference instrument. However, the paper is largely unfocused on a specific application reading as an unhappy hybrid of a measurement report (listing off communication protocols and daily readings), and letters paper (brief punchy analysis). As a result, the findings are muddled with distracting information more suitable to an Supplement. Furthermore, the analysis itself lacks impact.

**Reply**: The primary objective of this work is to investigate the performance of low-cost sensors (LCS) against a reference grade instrument across different sampling frequencies, and providing a better understanding of how to choose the sampling rate for LCS in high-pollution environments. Thus, the manuscript is not focused on a specific application but rather considers different deployment scenarios. Based on this feedback, we will aim to better articulate the focus of this work in the revised manuscript.

**Reviewer Comment 1.2** — The findings on LCS sensor accuracy as a function of averaging time has been investigated previously in literature - in North India no less (doi.org/10.5194/amt-11-4823-2018) however, here it lacks a nuanced discussion of aerosol optical characteristics and the influence of different sensor technology on the underlying quantification of plumes. It is unclear how these findings could be directly applied in a novel way for health studies, regulation, source apportionment, etc.

**Reply**: Thank you for this comment. While there has been prior studies on LCS accuracy as a function of sampling frequency, our study takes a different approach by focusing on the role of sampling frequency in determining the performance of LCS across different deployment scenarios.
In this work, our focus was to understand the performance of LCS which are typically factory calibrated using optical particle counters. However, we recognise that mentioning optical particle counters without further clarification may have lead to confusion, and in the revised manuscript we will explicitly state that our discussion of optical particle counters pertains to the calibration process of LCS rather than

aerosol optical characteristics.

The broader application of our findings, in the context of health studies, regulation, and source apportionment, would require careful sensor deployment strategies. The reason for mentioning these applications is to highlight that device power consumption is a key constraint when deploying LCS in the field. Since sampling frequency is directly linked to power consumption, understanding the trade-off between sampling resolution and power consumption is critical for ensuring long-term, sustainable sensor operation. We will revise the discussion section to better articulate the connection between sampling frequency, device power consumption, and real-world applications, to clarify the contributions of our work and differentiate it from previous studies. We appreciate the reviewer's feedback and we will make the necessary revisions to improve clarity.

**Reviewer Comment 1.3** — In addition to the underlying structural problems of the manuscript, the organization of the paper is weak. Nearly every figure requires regeneration or rethinking to meet the standards laid out by the publisher.

**Reply**: Thank you for your feedback. We will reconsider the organization of the manuscript and we will revise the figures based on the reviewer's specific comments. Additionally, we will ensure that the figures adhere to the color scheme recommended by the EGU and meet the required publication standards. We will revise the manuscript to enhance its clarity and presentation.

**Reviewer Comment 1.4** — Furthermore, literature review largely sticks to well cited more general work (e.g., Zimmerman et al) while not citing similar work from the region (e.g., `doi.org/10.3390/s20051347`). A more through analysis of work from Delhi and similar high background signal environments is sorely needed to better contextualize the results.

**Reply**: Thank you for your suggestion to include more region-specific literature, such as the study mentioned above (doi.org/10.3390/s20051347). We are aware of this work, however it primarily focuses on PM10 and compares LCS measurement with that from instruments such as SMPS, APS, and other OPS; while the work in this manuscript focuses on evaluating the impact of different sampling frequencies on LCS performance and their potential for different applications.

**Reviewer Comment 1.5** — With these criticisms in mind, as well as my more specific comments below, I recommend the paper be rejected as unsuitable for publication in AR at this time. I empathize with the authors as I fully understand the difficult and often grueling nature of field work and scientific writing. I believe the underlying data is useful. If the authors develop a new hypothesis focusing on a specific application (perhaps in coordination with the other instruments/historical data at IIT), the work could eventually lead to a useful reference for the broader community. I wish all the authors good luck.

**Reply**: Thank you for your valuable feedback. We strongly believe that we can fully address the reviewer's general and specific comments (discussed next) resulting in a significantly improved version of the manuscript.

**Specific Comments**

**Reviewer Comment 1.6** — The title is too vague. I recommend a hypothesis centered title, something like "Lower Cost PM Sensor Accurately Quantify Short-term Plumes in High Background Polluted Environments." This is more informative to readers and easier to find in search engines.

**Reply**: Thank you for your valuable feedback and suggestion regarding the title of the manuscript. We agree that a more hypothesis-centered title would better convey the focus of our study. We propose to revise the title to: "Impact of Sampling Frequency on Low-Cost PM Sensor Performance under Short-Term Temporal Events in High PM Environments". Thank you again for your comments and for helping us improve the clarity and quality of the manuscript.

**Reviewer Comment 1.7** — Citation formatting is incorrect. In-text citations should be in chronological order, and the references section should be organized alphabetically by last name. This may seem overly picky, however cross checking citations is key to the review process, and missing this step unnecessarily added time to my review. See other recent preprints (e.g., `ar. copernicus.org/preprints/ar-2024-35/`) for examples.

**Reply**: Thank you for pointing out the issues with the citation formatting. We sincerely apologize for the oversight and any inconvenience it may have caused during your review. We will carefully revise the in-text citations so they are in chronological order and reorganize the reference section alphabetically by last name. Your feedback is greatly appreciated, and we will correct these in the revised manuscript.

**Reviewer Comment 1.8** — In the introduction, please define what affordability means in the context of "low-cost" sensors. Are you using a benchmark for "low-cost" (e.g., compared to other non-regulatory optical monitors)? Furthermore, it maybe worth noting these are low capital cost sensors - clearly the work in deployment, and calibrating amounts to a very high cost.

**Reply**: Thank you for your suggestion. Currently, there is no guideline by the USEPA or other agencies defining the cost threshold for low-cost sensors, as pricing can be subjective and varies across countries or regions. In this study, "low-cost" refers to sensors that are significantly less expensive compared to reference-grade instruments. We will clarify this in the revised manuscript to emphasize that our use of the term "low-cost" is relative to the cost of reference grade instruments and not necessarily linked to a fixed price point.

**Reviewer Comment 1.9** — In the second paragraph of the introduction, you write LCS are compared to OPC. However, recent work (doi.org/10.1080/02786826.2023.2285935) has conclusively demonstrated LCS such as the Plantower cited this work are OPCs, albeit low efficiency OPCs. Please briefly clarify the differences in operating methodology between a reference OPC and an LCS OPC. Additionally, highlight how this difference could lead to mischaracterization of short term peaks. Especially since OPCs infer mass concentration from particle counts rather than directly measuring mass.

**Reply**: We appreciate the reviewer's suggestion. In the second paragraph of the introduction, our intention was to state that LCS are factory calibrated against higher-grade OPCs, as mentioned in their datasheets, rather than to imply a fundamental difference in sensor operating principle.

**Reviewer Comment 1.10** — The third paragraph references opinions or beliefs from the scientific community using language like "widely assumed" or "lack of understanding" but offers no literature or evidence. Please either support these statements with some theoretical backing in supplement, and/or cite (preferably recent) review work on the current understanding of high resolution monitoring.

**Reply**: Thank you for pointing out the need for supporting theoretical evidence or references. We will revise all such statements by adding appropriate recent references and, if necessary, include theoretical backing to strengthen the claims.

**Reviewer Comment 1.11** — Furthermore, in the third paragraph of the introduction it would help to be specific when using terms like "high frequency" and "low frequency". Techniques such as eddy covariance would consider 1 Hz or higher standard, while many epidemiological studies use annual or decadal data as standard. Only "high frequency" is defined earlier in the manuscript.

**Reply**: Thank you for your valuable suggestion. We agree that providing specific definitions and context for these terms is essential. In the revised manuscript, we will explain the standards for both high frequency and low frequency to ensure clarity.

**Reviewer Comment 1.12** — Again, in paragraph 4 of the introduction it is worth discussing what the LCS actually measures. For example, most PM LCS don't accurately identify resuspended dust from vehicles which can make up a significant portion of supramicron (dp > 1 micron) PM2.5 nor fresh vehicle emissions (dp < 0.1 microns). While the effective size range of most PM LCS is still useful, it largely lies in the accumulation mode, which includes a big share of the regional aerosol burden rather than the hyperlocal burden. Please rework this paragraph to discuss which short-term PM signals LCS are useful to quantify in the light of previous epidemiological work and aerosol fundamentals.

**Reply**: We appreciate the reviewer's insightful comment and agree that providing a discussion of LCS measurement range and capability will enhance the manuscript. As noted in previous studies, the LCS used in this work has a valid detection range for its first bin ($0.3 - 1.0$ $\mu$m) of approximately $< 0.9$ $\mu$m. The second, third, and fourth bins (nominally $1.0 - 2.5$, $2.5 - 4.0$, and $4.0 - 10$ $\mu$m) have nearly identical detection ranges of approximately $0.7 - 1.3$ $\mu$m. This suggests that these bins may have been factory-calibrated using the same test aerosol, limiting their ability to distinguish supramicron PM. We will revise paragraph 4 of the introduction to clarify the detection limitations of LCS and discuss which short-term PM signals LCS are suitable for quantifying, particularly in the context of epidemiological studies where accumulation-mode PM has been linked to long-term health effects. We appreciate this valuable suggestion and we will revise the paragraph to ensure a more accurate and complete discussion.

**Reviewer Comment 1.13** — The last paragraph of the introduction needs to be more specific on why this month was useful, why the location is valuable, and why the Sensiron sensor rather than the Plantower, or Honeywell monitor. Citing other work contrasting the Sensirion with other sensor models, especially in Delhi or similar environments, is useful.

**Reply**: Thank you for your feedback. In the revised manuscript, we will clearly explain why this month was particularly useful, why the location holds value, and why the Sensirion sensor was selected over

alternatives such as Plantower or Honeywell. We will also include relevant references contrasting the performance of Sensirion with other sensor models, especially in Delhi or similar environments. Your suggestions are greatly appreciated.

**Reviewer Comment 1.14** — There is a key strength of this manuscript which is excluded from the introduction and should be better described. This work focuses on characterizing plumes in highly polluted ambient environments. Much of the previous work in this area focus on comparing plumes in low to moderately polluted ambient environments in which the signal to background is high so analysis is straightforward. Despite the largely regional signal in Delhi, for which the design of many LCS is optimized, this work demonstrates its usefulness in source identification.

**Reply**: Thank you for your comment. In the revised manuscript, we will emphasize this aspect as well, clearly outlining how our study addresses the challenges of source identification in high-pollution environments such as Delhi, where the signal-to-background ratio is complex. We will also better describe the unique value of our findings in this context.

**Reviewer Comment 1.15** — Figure 1 should go to the Supplement, as should probably all of Section 2.2. These details are ubiquitous to most LCS packages, and are distracting in the main manuscript.

**Reply**: Thank you for your suggestion regarding Figure 1 and Section 2.2. We agree that some details in these sections may be ubiquitous to most LCS packages and can be streamlined. While we can move Figure 1 to the Supplement, we believe that entirely relocating Section 2.2 may result in loss of critical information that is relevant to the main manuscript. However, we will carefully revise this section to streamline the content, ensuring that only essential information is retained.

**Reviewer Comment 1.16** — I suggest adding 2 or at most 3 sentences in Section 2.3 contrasting how the BAM technique assess particle mass concentrations differently than an optical technique and why it matters for source identification.

**Reply**: Thank you for your valuable suggestion. We agree that contrasting the BAM technique with optical techniques in Section 2.3 would provide important context for understanding their differences and implications for source identification. In the revised manuscript, we will add a discussion explaining how the BAM technique assesses particle mass concentrations differently (e.g., through beta attenuation) compared to optical techniques (e.g., light scattering), and why these differences matter for accurately identifying and characterizing pollution sources.

**Reviewer Comment 1.17** — Lines 85-90 would be more efficiently communicated as a table with the 25th, 50th, and 75th percentiles for the relevant parameters.

**Reply**: Thank you for your suggestion. In the revised manuscript, we will include a table summarizing these percentiles to better communicate the data.

**Reviewer Comment 1.18** — Figure 2 should be more clearly illustrated. For PM2.5 and RH, please ensure the y-axis minimum is exactly 0 bounded. For all plots the y-axis minimums and maximums should be clearly marked. The x-axis date labels should be more clearly marked, Oct

29 and Nov 1 overlap in a confusing to read way. Consider excluding the year after the first date marker. Adding grid lines would also improve interpretation.

**Reply**:  Thank you for your detailed feedback on Figure 2. Your suggestions are greatly appreciated, and we will make these changes in the revised figure.

**Reviewer Comment 1.19** — Figure 4 needs to be entirely regenerated. The top panel (a), should not contain both reasonably 0 bounded elements (i.e., PM2.5 and RH) on the same axis as temperature. Additionally, please label the beginning and end points on x-axis. Furthermore, when comparing different colored lines on the same plot, it is best practice to use color-blind friendly color schemes (see EGU publishing guidelines for more details) and/or different line styles. In this case, I'd recommend 3 different subplots for each variable. Panel (b), as in Figure 2 should have clear bounds, and be 0 bound on the y-axis minimum. Also, the red text in panel b should be better spaced to avoid overlap with the highlighted boxes or other text. Panel (c) looks to me like a screenshot from Plotly. I strongly recommend against Plotly and similar interactive tools for publication since they are difficult to reproduce. Also the y-axis label in Panel c is insufficient (no units!). Please remove Panel (c), or replace with a clearly labeled static boxplot, or use a table to describe the key percentiles of the monthly distribution.

**Reply**:  Thank you for your detailed feedback on Figure 4. We acknowledge the issues you have highlighted and agree that the figure requires improvement. Regarding panel (a), while temperature is not 0-bound, the lowest temperature in our study was above 16°C, which is why we kept temperature together with PM2.5 and RH. We will label all axes clearly, use color-blind-friendly schemes, and improve panel (b) by clearly bounding the y-axis and spacing the red text. Panel (c) will be replaced with a static box-plot or a table summarizing key percentiles.

**Reviewer Comment 1.20** — Figure 5 needs to be entirely regenerated. All scatterplots should be "square" - or equal ranges on both the x- and y-axes. The long lists of metrics would be easier to read on an accompanying table, with the panel labels in the upper left corner of the plot. Additionally, a legend denoting what each line and scatter point corresponds to is compulsory. To my eye, it appears as though there is some red shading around the regression line. Please explain how this is derived and how it is relevant. If not relevant, remove. I would guess its a confidence interval based on the sample size and therefore irrelevant and distracting.

**Reply**:  Thank you for your detailed feedback on Figure 5. We agree that the figure requires improvements for better clarity and presentation. In the revised manuscript, we will regenerate the scatter plots with equal ranges on both the x- and y-axes. We will also include a legend inside each figure to denote what each line and scatter point corresponds to. Additionally, the red shading around the regression line, which represents a confidence interval, will be removed as it is not directly relevant to the analysis. We appreciate your suggestions and we will make these changes in the revised manuscript.

**Reviewer Comment 1.21** — Figure 6 needs to be entirely rethought. As it stands the figure is too low resolution to read normally, I had to zoom in to read it and it rendered blurry. The "clock plots" don't follow best practices as highlighted in the previous figure comments, and don't add anything not clearly visible from the time series plots. Focusing on a few episodes can be helpful, especially as these demonstrate clear contrast, but the takeaway seems uninteresting.

**Reply**:  Thank you for your feedback on Figure 6. We will upload a higher-resolution version in the revised manuscript to address any concerns about clarity. Regarding the clock plots, we believe they provide a unique visualization of diurnal patterns, which complements the time series plots. However, we will reconsider their presentation to align with best practices and ensure they add clear value. We will also highlight the importance of these plots in the revised manuscript to better convey their relevance to the analysis.

**Reviewer Comment 1.22** — Figure 7 is also too low resolution (i.e., blurry) and should use colorblind friendly perceptually uniform colormaps in accordance with EGU publishing guidelines. Furthermore, there should be some discussion of aerosol properties which explain the "swings" in LCS hourly and sub-hourly data. Clearly the LCS is also systemically underestimating in addition to overestimating (probably due to RH?). Therefore, the magnitude of the plume spikes maybe incorrectly quantified relative to the BAM. Some baseline-spike decomposition algorithm could more holistically address these limitations. There are many in literature, especially in mobile monitoring.

**Reply**:  Thank you for your feedback on Figure 7. We will upload a higher-resolution version and also update the figure to use colorblind-friendly scheme. Additionally, we will include a discussion on aerosol properties to explain the observed swings in LCS hourly and sub-hourly data, as well as address the systemic underestimation and overestimation issues, potentially linked to RH. Your suggestions are greatly appreciated, and we will make these improvements in the revised manuscript.

**Reviewer Comment 1.23** — Overall the plume discussion doesn't focus on investigating aerosol characteristics. Simply finding a spike does not necessarily offer useful or actionable information. Therefore topics of interest to the community including background-plume decomposition analysis - which is key to both epidemiology and source characterization - are not clearly discussed or analyzed in the conclusion.

**Reply**:  Thank you for your insightful comment regarding the plume discussion. We acknowledge that the current manuscript primarily focuses on investigating the capability of LCS to capture high plume events and the impact of sampling frequency on their performance relative to the reference BAM. While the goal was not to provide actionable guidelines for plume events, we agree that a more detailed discussion on how this data can be useful for epidemiology and source characterization would add value. In the revised manuscript, we will expand the discussion to include potential applications of the data for background-plume decomposition analysis and its relevance to epidemiology.

**Technical Comments**

**Reviewer Comment 1.24** — Line 65 - Needs a citation (maybe more than 1) when you assert its "among the best"

**Reply**:  Thank you for pointing this out. We will include relevant citations in the revised manuscript to substantiate this claim, drawing from recent literature that evaluates the performance of low-cost sensors, including the Sensirion model, in similar environments.

**Reviewer Comment 1.25** — Line 66 - It is customary to refer to Supplement items as Table S1, Figure S1, or Section S1. Refering to the sensors as S1, S2, etc. could be confusing for readers in publication.

**Reply**: Thank you for pointing this out. To avoid any ambiguity, in the revised manuscript, we will use a different labeling convention for the sensors, such as Sensor 1, Sensor 2, etc.

**Reviewer Comment 1.26** — Lines 87-88 - This is not PM2.5 exposure, its simply the mean mass concentration.

**Reply**: Thank you for catching this inaccuracy. The term "PM2.5 exposure" is not appropriate here, as it refers to mean mass concentration. We will revise the text to accurately reflect this by replacing "PM2.5 exposure" with "mean PM2.5 mass concentration".

**Reviewer Comment 1.27** — Line 101 - While higher temperatures can result in lower PM2.5 concentrations, I think this is merely an indicator of increased ventilation due to higher planetary boundary layer height during midday rather than a direct temperature effect - please clarify.

**Reply**: Thank you for your insightful comment regarding the relationship between temperature and PM2.5 concentrations. We will clarify this point in the revised manuscript to better reflect the underlying meteorological dynamics.

**Reviewer Comment 1.28** — Line 112 - This looks like the standard way to calculate SD, please either simply cite a statistics textbook/manual or move to supplement.

**Reply**: Thank you for your comment. We will address this in the revised manuscript as per your suggestion.

**Reviewer Comment 1.29** — Line 118 - Since you're using US-EPA standards, please state the US-EPA guideline for CV (about 0.1 or 10%, although 0.2 or 20% is a commonly used more lax requirement for LCS).

**Reply**: Thank you for pointing out the need to clarify the USEPA guideline for the coefficient of variation (CV). We will add this in the revised manuscript.

**Reviewer Comment 1.30** — Line 125 - Please explain or cite why this is the case

**Reply**: Thank you for your comment. We will provide an explanation or include a citation to support the statement, ensuring that the reasoning or evidence behind this claim is clearly indicated.

**Reviewer Comment 1.31** — Overall the plume discussion doesn't focus on investigating aerosol characteristics. Simply finding a spike does not necessarily offer useful or actionable information. Therefore topics of interest to the community including background-plume decomposition analysis - which is key to both epidemiology and source characterization - are not clearly discussed or analyzed in the conclusion.

**Reply**: Thank you for your feedback. We would like to clarify that the primary objective of this work and the manuscript is to investigate the performance of low-cost sensors (LCS) against a reference grade

instrument across different sampling frequencies, and providing a better understanding of how to choose the sampling rate for LCS in high-pollution environments. While we acknowledge the importance of background-plume decomposition analysis for source characterization and epidemiology, this study is not intended to serve as a guideline for these applications. Instead, it focuses on evaluating the capability of LCS to capture high-frequency variations, such as plume events, relative to the reference BAM. We appreciate your insights and we will articulate the focus of this work clearly in the revised manuscript.

---

## Author Comment (AC2)

**Response to Reviewer 2 Comments on Manuscript ar-2024-39 "Impact of Sampling Frequency on Low-Cost PM Sensor Performance"**

The authors would like to thank the editor and reviewers for their valuable feedback on the manuscript. In this document, we present our responses to the reviewer comments and suitable changes will be made in the revised version of the manuscript addressing these comments. For the reviewers' convenience, the reviewer comments are shown in **black**, and our response to these comments are shown in **blue**

**Reviewer 2**

**Reviewer 2.1** — The paper presents a field study in which a Low Cost Sensor measurement station for PM2.5 is designed and operated during one month on the roof of a building of Indian Institute of Technology (New Delhi campus). The data are analyzed and compared to reference measurement obtained by BAM Beta attenuation mass monitor thank to different sampling frequencies by the Low Cost Sensor Station. The general context of the study is interesting, it deals with configuration of sampling frequency of Low Cost Sensors regarding power consumption especially for remote deployments and what is it possible to characterize with in term of short pollution event. The precise objectives of the paper are clearly described. The paper is well written, and results are clearly presented. It is in line with topics of Aerosol Research. Nevertheless, some important points have to be accounted to improve the paper and avoid any misinterpretation.

**Reply**: Thank you for your summary and valuable feedback. We address your detailed comments below.

**General Comments**

**Reviewer 2.2** — The main comment I have on the paper is to clarify the definition of the sampling frequency/sampling interval and related discussion on the effect of this parameter on the results. It is not clear to what correspond exactly LCS sampling frequencies named 5, 10, 15, 30, 60 min and how they are obtained.

**Reply**: Thank you for your comment regarding the clarification of sampling frequency and its impact on the results. To clarify, the LCS data is collected/sampled every 15 seconds. The sampling intervals of 5, 10, 15, 30, and 60 minutes refer to the time intervals at which a single sample was extracted from the midpoint of each duration. For example, 5-minute sampling means that if a sample was taken at 2.5 minutes, the next one would be taken at 7.5 minutes, and so on. Hourly averages for each sampling interval were calculated by averaging all the samples collected within that hour. We will ensure this explanation is clearly and concisely included in the revised manuscript.

**Reviewer 2.3** — As it is written it let thinking that data corresponding to such frequencies are obtained by doing periodic average on the raw measurements done by LCS working at an effective sampling frequency of 15 seconds. This means that sampling frequency of the LCS is not changed

during experiments. This as to be clarified in the paper and the title of the paper should be adapted. In fact, if the frequency studied by the authors is a periodic average obtained by post-treatment it has no relationship with LCS intrinsic performance. The title should avoid such misunderstanding.

**Reply**: Thank you for your comment. During the experiment, the LCS is operated at a fixed sampling interval of 15 seconds, and the data for the other sampling intervals (5, 10, 15, 30, and 60 minutes) is obtained by taking the sample at the middle of each sampling period. This procedure was adopted only for the ease of carrying out the experiment and the analysis across sampling frequencies. We will clarify this in the revised manuscript to avoid any misunderstanding.

**Reviewer 2.4** — The authors should improve the paper by better describing how the LCS data are acquired: if it is always active sampling during one month of if there sleep mode periods between measurements periods?

**Reply**: Thank you for the comment. The low-cost sensors were continuously sampling throughout the experiment, with no sleep modes implemented during the measurement period. We will revise the manuscript to explicitly describe the data acquisition process to ensure clarity.

**Specific comments**

**Reviewer 2.5** — Page 3, line 80 Precise/confirm that BAM unit is equipped with PM10 Inlet + PM2.5 Cyclone (which model VSCC or URG?)

**Reply**: Thank you for the comment. The BAM unit used in this study is equipped with a PM2.5 Very Sharp Cut Cyclone (VSCC). We will add this information to the manuscript for better clarity.

**Reviewer 2.6** — Page 5, lines 106-107 Give additional information to explain the difference between data aggregated on 60 min interval and the hourly average.

**Reply**: Thank you for your comment. The 60-minute sampling interval refers to a single sample taken at the midpoint of each hour (e.g., at 30 minutes), while the hourly average is calculated by averaging all samples collected within that hour (e.g., all the 15-second samples from 00:00 to 00:59). We will include this explanation in the revised manuscript for better clarity.

**Reviewer 2.7** — Page 11, fig. 7 Improve readability of titles

**Reply**: Thank you for your feedback. We will revise Figure 7 by improving the clarity, font size, and formatting of the titles to ensure they are legible and visually consistent.

**Reviewer 2.8** — Page 11, lines 168-171 The conclusion of the paper should be adapted to avoid misunderstanding about energy consumption minimization of LCS according to finding of this study. Energy consumption is not directly studied here and no evidence are given that operation of LCS with lower energy consumption due to lower effective sampling frequency provide comparable measurements.

**Reply**: Thank you for the feedback. To address this concern, we will include data that directly correlates sampling frequency with energy consumption, helping readers better understand the relationship between

these parameters. Additionally, we will revise the conclusion to clearly state our findings regarding energy consumption based on the available data.

---

## Referee Report (RR1)

**Comments on Revised Manuscript ar-2024-39**

**Title: Impact of Sampling Frequency on Low-Cost PM Sensor Performance including short-term temporal events in high PM environments**

The revised manuscript was greatly improved thanks to reviewer comments with better clarity and suitable additional information about power consumption.

Some minor points should be accounted for to give the reader the best understanding of the results.

**Page 6, § 3**
The authors should give some more details about the comparison between the five LCS measurements to justify further analysis based on only one unit. In particular:
- Is the CV=4.38 % corresponding to just one daily average comparison is representative of other daily periods?
- Does the US EPA standards mentioned: SD< 5 µg/m$^3$ and CV< 30 % are especially related to intercomparison between same instruments? Does the CV<30% should be related to only one daily measurement serie or to a 30 days serie?

Give the precise reference for the US EPA standard in the References list P14:
Duvall, R., A. Clements, G. Hagler, A. Kamal, Vasu Kilaru, L. Goodman, S. Frederick, K. Johnson Barkjohn, I. VonWald, D. Greene, AND T. Dye. Performance Testing Protocols, Metrics, and Target Values for Fine Particulate Matter Air Sensors: Use in Ambient, Outdoor, Fixed Site, Non-Regulatory Supplemental and Informational Monitoring Applications. U.S. EPA Office of Research and Development, Washington, DC, EPA/600/R-20/280, 2021.

**Page 8 §3.1**
Give definition for MAE

**Page 9, figure 4**
I understand, figure 4 corresponds to an average of a define number of 15 s measurement taken at the mid-point of the sampling frequency. So each point fig 4a corresponds to an average of 240 values, fig 4b average of 12 values, fig 4c average of 6 values, fig 4d average of 4 values, fig 4e average of 2 values and fig 5c single value.
Surprisingly the data are not more scattered when single or two values are considered for the hourly average. Could the authors add more comment on that in the text § 3.1. May be also starting to introduce fig 5 in § 3.1 showing the 15 s raw data on a one-hour serie with the different averaged values.

**Page 12, line 205**
Recall that the value 4.38% corresponds to coefficient of variation between five SPS30 units on single daily analysis.

**Page 12, line 209**
Precise "minimal impact on the hourly measurement accuracy"

---

## Author Response (AR2)

Authors' Response to Reviewers' Comments
Manuscript ID# ar-2024-39
Revised Title: "Impact of Sampling Frequency on Low-Cost PM Sensor Performance including Short-Term Temporal Events in High PM Environments"
Original Title: "Impact of Sampling Frequency on Low-Cost PM Sensor Performance"

The authors would like to thank the editor and reviewers for their valuable feedback on the manuscript. We have addressed the comments which has considerably improved the quality of the manuscript. Our responses to the comments and suggestions are provided in this document, and suitable changes have been incorporated in the revised version of the manuscript. For the reviewer's convenience, the reviewer comments and our response to the reviewer comments are shown in **black** color font and the modified parts of the manuscript are shown in **blue** color font.
* * *
**Reviewer 1**

**General Comment**

The revised manuscript was greatly improved thanks to reviewer comments with better clarity and suitable additional information about power consumption. Some minor points should be accounted for to give the reader the best understanding of the results.

**Response**: The authors thank the reviewer for the feedback and valuable suggestions which allowed us to revise the manuscript and improve its overall quality.

**Comment 1.1** — Page 6, § 3
The authors should give some more details about the comparison between the five LCS measurements to justify further analysis based on only one unit. In particular:
  • Is the CV=4.38% corresponding to just one daily average comparison is representative of other daily periods?
  • Does the US EPA standards mentioned: SD$\leq$ 5 µg/m3 and CV$\leq$ 30% are especially related to intercomparison between same instruments? Does the CV$\leq$30% should be related to only one daily measurement serie or to a 30 days serie?

**Response**: Thank you for the comment. The coefficient of variation (CV = 4.38% ) was calculated using daily averages of 15-second data over the 30-day period and is not a single day average. This ensures that the reported CV value reflects long-term consistency across all five LCS units. The low value of CV indicates strong inter-sensor agreement, validating the use of a single representative unit for further analysis. The cited US EPA precision targets (SD $\leq$ 5 µg/m³ and CV $\leq$ 30%) are applicable for intercomparison between identical instruments over a 30-day period, as specified in:

- Duvall et al. (2021), Section 3.1.3: Precision metrics for collocated sensors.

- Duvall et al. (2021), Section 2.1: Requirement for 30-day evaluation to assess sensor precision.

Our analysis adhered to these guidelines, with collocated data collected over 30 days to compute SD and CV. As suggested we have now revised the 1st paragraph of Section 3 to improve clarity. The revised part of Section 3 is reproduced below:

"Daily averages for all the LCS units were computed using 15 second resolution data over the 30 day study period. This month-long dataset allows for diverse ambient exposure conditions and diurnal variations. During the study, hourly averaged reference BAM measurements ranged from 11 to 303 $\mu$g/m$^3$, with a mean value of 84.41 $\mu$g/m$^3$. Despite this wide PM$_{2.5}$ concentration range, the calculated standard deviation (SD) of 3.92 $\mu$g/m$^3$ (below the USEPA limit of $< 5$ $\mu$g/m$^3$) and coefficient of variation (CV) of 4.38% (significantly lower than the USEPA limit of $< 30\%$) indicates high precision among the five LCS units (Duvall et al., 2021). Given this high level of precision, a single LCS unit (Sensirion-1) was used for the subsequent analysis, as repeating the analysis with the other LCS units would yield similar results."

**Comment 1.2** — Give the precise reference for the USEPA standard in the References list P14: Duvall, R., A. Clements, G. Hagler, A. Kamal, Vasu Kilaru, L. Goodman, S. Frederick, K. Johnson Barkjohn, I. VonWald, D. Greene, AND T. Dye. Performance Testing Protocols, Metrics, and Target Values for Fine Particulate Matter Air Sensors: Use in Ambient, Outdoor, Fixed Site, Non-Regulatory Supplemental and Informational Monitoring Applications. U.S. EPA Office of Research and Development, Washington, DC, EPA/600/R-20/280, 2021.

**Response**: Thank you for the feedback. In the revised manuscript, this reference has been corrected.

**Comment 1.3** — Page 8 §3.1
Give definition for MAE

**Response**: Thank you for the feedback. In the revised manuscript, MAE has now been included in Section 3.1 alongside the other performance metrics. For a detailed description of MAE and other performance metrics, we have cited the work of Zimmerman (2022). The revised part of Section 3.1 is reproduced below:

"To assess the impact of LCS sampling interval on the correlation of the LCS data with BAM, we compute the coefficient of determination ($R^2$), slope (m), intercept (b), mean absolute error (MAE), root mean square error (RMSE), and normalized root mean square error (NRMSE) (Duvall et al., 2021; Zimmerman, 2022)."

**Comment 1.4** — Page 9, figure 4

I understand, figure 4 corresponds to an average of a define number of 15 s measurement taken at the mid-point of the sampling frequency. So each point fig 4a corresponds to an average of 240 values, fig 4b average of 12 values, fig 4c average of 6 values, fig 4d average of 4 values, fig 4e average of 2 values and fig 5c single value. Surprisingly the data are not more scattered when single or two values are considered for the hourly average. Could the authors add more comment on that in the text § 3.1. May be also starting to introduce fig 5 in § 3.1 showing the 15 s raw data on a one-hour serie with the different averaged values.

**Response**: We thank the reviewer for this insightful observation and comment. We would like to clarify that the reference BAM data in our study is available only at hourly resolution. Thus, in Figure 4, each subplot compares hourly averaged LCS measurements (derived from LCS measurements at different sampling frequencies) against the corresponding hourly BAM values. Regardless of the LCS sampling interval (15-second, 5-minute, etc.), the LCS data in that hour is aggregated to hourly average before comparing with the BAM data. So, each subplot in Figure 4 contains 720 hourly points (30 days $\times$ 24 hours). As the temporal resolution of comparison (hourly) remains the same for each subplot the scatter does not increase with different sampling intervals for LCS data. To clarify this, we have revised Section 3.1 and the revised part is reproduced below:

"Each subplot in Figure 4 corresponds to a different sampling interval (e.g., 15 minutes, 30 minutes, etc.), where hourly LCS averages are computed from the raw measurements within that hour. Despite the different number of data points per hour (e.g., four data points for 15 minutes interval, two data points for 30 minutes interval), each subplot contains 720 hourly points (30 days $\times$ 24 hours), resulting in the same temporal resolution."

**Comment 1.5** — Page 12, line 205

Recall that the value 4.38% corresponds to coefficient of variation between five SPS30 units on single daily analysis.

**Response**: We have modified the concerned sentence in the revised manuscript for better clarity. The revised sentence from Section 4 is reproduced below:

"The SPS30 LCS units demonstrated high precision, with a CV of $4.38\%$ during the 30 days study period."

**Comment 1.6** — Page 12, line 209

Precise "minimal impact on the hourly measurement accuracy

**Response**: Thank you for this comment. We have now revised the referred sentence from Section 4 which is reproduced below:

"In addition, our analysis revealed that varying the sampling frequency had minimal impact on the hourly measurement accuracy, however only high frequency sampling ($15$ seconds sampling interval) was effective in capturing the transient plume events."